ADHDP-based robust self-learning 3D trajectory tracking control for underactuated UUVs

Zhao Chunbo
http://orcid.org/0000-0001-5401-895X Yan Huaran huaranyan202202@163.com
Gao Deyi
Merchant Marine College, Shanghai Maritime University , Shanghai , China
Balas Valentina Emilia
Electronic publication date: 2024 Dec 10
Publication date: 2024
Volume: 10
Electronic Location ID: e2605
Received 2024 May 31; Accepted 2024 Nov 21
Copyright: © 2024 Zhao et al.
Copyright year: 2024
Copyright holder: Zhao et al.
License: This is an open access article distributed under the terms of the Creative Commons Attribution License, which permits unrestricted use, distribution, reproduction and adaptation in any medium and for any purpose provided that it is properly attributed. For attribution, the original author(s), title, publication source (PeerJ Computer Science) and either DOI or URL of the article must be cited.
License URL: https://creativecommons.org/licenses/by/4.0/

Keywords: Unmanned underactuated vehicles (UUVs), Robust adaptive control, Trajectory tracking, Action-dependent heuristic dynamic programming (ADHDP)

Funding: The authors received no funding for this work.

==============================
In this work, we propose a robust self-learning control scheme based on action-dependent heuristic dynamic programming (ADHDP) to tackle the 3D trajectory tracking control problem of underactuated uncrewed underwater vehicles (UUVs) with uncertain dynamics and time-varying ocean disturbances. Initially, the radial basis function neural network is introduced to convert the compound uncertain element, comprising uncertain dynamics and time-varying ocean disturbances, into a linear parametric form with just one unknown parameter. Then, to improve the tracking performance of the UUVs trajectory tracking closed-loop control system, an actor-critic neural network structure based on ADHDP technology is introduced to adaptively adjust the weights of the action-critic network, optimizing the performance index function. Finally, an ADHDP-based robust self-learning control scheme is constructed, which makes the UUVs closed-loop system have good robustness and control performance. The theoretical analysis demonstrates that all signals in the UUVs trajectory tracking closed-loop control system are bounded. The simulation results for the UUVs validate the effectiveness of the proposed control scheme.

Introduction

Recently, uncrewed underwater vehicles (UUVs) have garnered substantial interest from research teams due to potential engineering applications such as underwater archaeology, torpedo deployment, and cable installation (Peng, Wang & Han, 2019; Qiao & Zhang, 2020; Wang et al., 2021a; Wang, Gao & Zhang, 2021; Yuan et al., 2023). Most oceanic missions require the tracking of a predefined desired trajectory. In practice, UUVs are often underactuated, displaying significant nonlinearity, model parameter uncertainties, and vulnerability to unknown time-varying disturbances in the ocean environment (Li et al., 2023; Zhou et al., 2023). These features greatly complicate and challenge the development of trajectory tracking controllers.

Since the majority of UUVs are outfitted with three propellers to control surge, pitch, and yaw motions, they are classified as underactuated systems, where the number of independent actuators is less than the degrees of freedom. Yet, the existence of underactuated characteristic makes the design of UUV trajectory tracking control law more difficult. Further, the incomplete development of the hydraulic system complicates the acquisition of UUV model parameters. The neural networks (NNs) have gained substantial attention in research because of its universal approximation property. In Shi et al. (2024), Van (2019), Liu & Du (2021), Wang et al. (2019), NNs were applied to approximate the unknown nonlinear dynamics. Nevertheless, achieving precise estimation with NNs necessitates weight identification, which adds considerable computational complexity. To overcome this issue, Shen et al. (2020) proposes combining minimum learning parameter technology with RBFNNs and a fixed-time sliding mode control scheme with RBFNN disturbance observer and the single-parameter learning was proposed in Zhu et al. (2023). A robust trajectory tracking control scheme is proposed in Heshmati-Alamdari, Nikou & Dimarogonas (2021) for underactuated underwater vehicle with obstacles in the constrained workspace. In Chen et al. (2017), a robust control method is developed though based on sliding mode control and fuzzy control for underactuated underwater vehicle with uncertain model parameters and unknown disturbances. Accurate trajectory tracking with good robustness is achieved based on a finite-time extended state observer in Wang et al. (2021b).

Currently, most proposed control schemes focus on achieving stable tracking without considering the optimal performance of UUVs under uncertain dynamics and time-varying disturbances. Nonetheless, the intricate marine environment inevitably impacts the control performance of systems with fixed design parameters. In recent times, there has been a wealth of research on the application of adaptive dynamic programming (ADP) in numerous systems to achieve optimal control (Tong, Sun & Sui, 2018; Li, Sun & Tong, 2019). The fault-tolerant tracking control problem for UUVs subjected to time-varying ocean current disturbance is converted into an optimal control problem through the ADP approach in Che & Yu (2020). Through the integration of the backstepping design method, a fault-tolerant control scheme is formulated using a single-critic network-based ADP approach in Che (2022). In Wen et al. (2019), virtual and actual control are designed as the optimization solution of corresponding subsystems, and an optimized backstepping technique based on an actor network is utilized to enhance the control performance of the surface vessel system. A fault-tolerant fuzzy-optimized control scheme for tracking control of underactuated autonomous underwater vehicles is proposed in Gong, Er & Liu (2024) for complicated oceanic environments, particularly with the existence of unknown actuator failures and uncertain dynamics. Based on the above analyses, it is imperative to explore an intelligent control scheme with self-learning and self-adaptive capability to improve the control performance of UUVs under uncertain dynamics and time-varying ocean disturbances.

Motivated by the above discussions, this work develops a robust self-learning control scheme based on action-dependent heuristic dynamic programming (ADHDP) for underactuated UUVs subject to uncertain dynamics and unknown time-varying external disturbances. The main contributions of this work are as follows. The fact that a large number of control design parameters (e.g., adaptive gain, leakage term gain, etc.) must be artificially set to fixed values makes it difficult to ensure the optimization of control performance metrics for existing robust adaptive control methods in Li et al. (2021) and Yang et al. (2021). The ADHDP-based control scheme proposed in this article can automatically update the control parameters in response to dynamic uncertainties and unknown time-varying perturbations, and our work not only reduces the need for manually designing the control design parameters, but also greatly improves the control performance and adaptability of the system.

Unlike traditional optimal tracking control schemes, such as those in Zhang, Li & Liu (2018) and Zheng et al. (2020), which usually require partial model information (e.g., the control gain is known or partially known) for parameter tuning and strategy optimization, the ADHDP-based control scheme proposed in this article no longer relies on the model parameter information, but only the input and output data of the UUV.

The remaining segments of this work are organized as follows. The problem formulation and preliminaries are introduced in “Problem Formulation and Preliminaries”. The details of design process and stability proof of the robust self-learning are given in “Control Law Design”. “Simulation Results” presents the simulation results. The conclusions of this work are summed up in “Conclusion”.

Notations: In this work, diag(∗) denotes the diagonal matrix; λ(∗) denotes the minimum eigenvalue of the matrix; ||∗|| denotes the 2-norm value of the matrix or vector; s(∗) and c(∗) denote the sine and cosine functions, respectively.

Problem formulation and preliminaries

Motion mathematical model of UUVs

The mathematical model of underactuated UUVs (Yan et al., 2024) is given as:

(1) η˙=J(η)v

(2) Mv˙+C(v)v+Dv+g(η)=τ+d

where η=[x,y,z,θ,φ]T is position vector consist of the surge position x, the sway position y, the heave position z, the pitch angle θ and the yaw angle φ of UUVs in the earth-fixed frame. v=[u,ν,w,q,r]T is the velocity vector consist of the surge velocity u, the sway velocity ν, the heave velocity w, the pitch velocity q and the yaw velocity r of UUVs in the body-fixed frame. In this article, we define the earth-fixed and the body-fixed reference frames of the UUV as indicated in Fig. 1. D=diag(d11,d22,d33,d44,d55) represents the hydrodynamic damping matrix. M=diag(m11,m22,m33,m44,m55) is the positive-definite inertia matrix. τ=[τu,0,0,τq,τr]T. is the control input vector consist of surge force τu, pitch moment τq and yaw moment τr. d=[d1,d2,d3,d4,d5]T represents the environmental disturbance vector. g(η)=[0,0,0,ρg∇GMLs(θ),0]T with ρ, g, ∇ and GML are water density, gravity acceleration, displaced volume and longitudinal metacentric height, respectively. J(η)=[(θ)c(φ)−s(φ)s(θ)c(φ)00c(θ)s(φ)c(φ)s(θ)s(φ)00−s(θ)0c(φ)000001000001c(φ)] represents the rotation matrix. C(v)=[000m33w−m22ν0000m11u000−m11u0−m33w0m11u00m22ν−m11u000] represents the Coriolis and centripetal force matrix.

Figure 1 The earth-fixed and the body-fixed reference frames of the UUV.

Assumption 1 The model parameters matrices M, C and D are unavailable.

Assumption 2 The disturbance vector d is bound yet unknown time-varying and ∥d∥≤dm, dm is a positive constant.

Assumption 3 The desired trajectory of UUVs in this article is notated as ηd=[xd,yd,zd]T. The component of ηd and their first two time derivatives are bound.

Assumption 4 The sway velocity ν and the heave velocity w are bounded.

Transformation of UUV’s position

To initiate the discussion, the subsequent coordinate transformation is employed to tackle the underactuation issue of UUVs:

(3) η1=[x+Fc(θ)c(φ),y+Fc(θ)s(φ),z−Fs(φ)]T

where F is a small positive constant.

According to Eqs. (1)–(3), we can obtain the following equation as:

(4) η˙1=J1(η)v1+R1(η,v)

(5) M1v˙1+C1(v)v1+D1v1+g1(η)=τ1+d1

where J1(η)=[(θ)c(φ)−Fs(θ)c(φ)−Fs(φ)c(θ)s(φ)−Fs(θ)s(φ)Fc(φ)−s(θ)−Fc(θ)0], M1=diag(m11,m44,m55), v1=[u,q,r]T, C1(v)=[0m33w−m22ν(m11−m33)w00(m22−m11)ν00], R1(η,v)=[−νs(φ)+ws(θ)c(φ)νc(φ)+ws(θ)c(φ)wc(θ)], τ1=[τu,τq,τr]T, d1=[d1,d4,d5]T, D1=diag(d11,d44,d55) and g1(η)=[0,ρg∇GMLs(θ),0]T.

The goal of this work is to propose a robust self-learning control law based on ADHDP to ensure that the UUV trajectory tracking error converges to a small compact set in the presence of uncertain dynamics, and unknown time-varying disturbances, while all signals in the UUV trajectory tracking closed-loop control system are bounded.

Lemma 1: (Luo et al., 2020; Na et al., 2020) Notate f(s) is a continuous function which is defined on a compact set s⊂Rn. There has a radial basis function neural network (RBFNN) for an arbitrarily small constant Θ satisfy

(6) f(s)=ω∗TG(s)+Θ

where ω∗ denotes optimal weight vector. Θ denotes the reconstruction error of NNs, which satisfies ||Θ||≤Θ¯. G(s)=[G1(s),G2(s),⋅⋅⋅,Gn(s)]T denotes the basic function vector. Herein, Gn(s) is chosen as the Gaussian function.

Control law design

Robust adaptive NN control law design

Notate the tracking error s1=[s11,s12,s13]T as:

(7) s1=η1−ηd.

According to Eqs. (5) and (7), we have

(8) s˙1=J1(η)v1+R1(η,v)+η˙d.

Design the virtual control law ς∈R3 for v1 as follows

(9) ς=J1−1(η)(−k1s1−R1(η,v)+η˙d)

where k1∈R3×3 is positive-definite design matrix.

To proceed, we can define the velocity error vector s2=[s21,s22,s23]T as follows:

(10) s2=v1−ς

In the light of Eqs. (6) and (10), we can obtain the time derivative of s2

(11) M1s2.=−C1(v)v1−D1v1−g1(η)+τ1+d1−M1ς.

Using the RBF NN to approximate the uncertain term −C1(v)v1−D1v1−g1(η)−M1ς.. We have

(12) −C1(v)v1−D1v1−g1(η)−M1ς.=ω∗TG(J)+Θ

where ω∗ is the ideal weight matrix, J=[vT,ςT,θ]T denotes the input vector, G(J) represents the basis function vector. Θ is the NN reconstruction error vector with ||Θ||≤Θ¯.

With the aid of virtual parameter learning technology, the following inequality can be obtained

(13) ∥ω∗TG(J)+Θ+d1∥≤ωm∥G(J)∥+Θ¯+dm≤Hϕ

where H=max{ωm,Θ¯+dm} is an virtual parameter without physical meaning. ϕ=∥GJ∥+1 is a scalar function.

Design the control law for control input as:

(14) τ1=−k2s2−J1(η)s1−H^ϕ2s2

where s2∈R3×3 denotes positive-definite design matrix, H^ denotes the estimate of H with estimation error H~=H−H^.

The adaptive law is given as

(15) H^˙=Γ(ϕ2||s2||2−ϑH^)

where Γ and ϑ are positive design constants.

Theorem 1 Consider the UUV trajectory tracking closed-loop control system Eqs. (1), (2) under Assumptions 1–4, the virtual control law Eq. (9), the control law Eq. (14), with the adaptive law Eq. (15). The actual trajectory η1 can track the desired trajectory ηd, while all signals in the UUV trajectory tracking closed-loop control system are bounded.

Proof: Select the following Lyapunov function candidate

(16) V=12(s1Ts1+s2TM1s2+Γ−1H~2)

The time derivative of Eq. (16) is

(17) V˙=s1Ts˙1+s2TM1s˙2+Γ−1H~(−H^˙)

In view of Eqs. (8)–(10), we have

(18) s1Ts1.=s1TJ1(η)s2−s1Tk1s1

Substituting Eqs. (13) and (14) into Eq. (11) and rearranging Eq. (11) yields

(19) s2TM1s˙2≤∥s2∥Hϕ+s2T(−k2s2−J1(η)s1−H^ϕ2s2)≤H4+s2T(−k2s2−J1(η)s1−H~ϕ2s2)

In view of Eq. (15), one has:

(20) Γ−1H~(−H^˙)=−H~(ϕ2∥s2∥2−ϑH^)≤−H~ϕ2||s2||2−12ϑH~2+12ϑH2.

Substituting Eqs. (18)–(20) into (17) and rearranging Eq. (17) yields:

(21) V˙= s1Ts˙1+s2TM1s˙2+Γ−1H~(−H^˙)≤−s1Tk1s1−s2Tk2s2−12ϑH~2+12ϑH2+H4≤−2σV+ψ

where σ=min{λmin(k1),λmin(k2M¯1−1),12ϑΓ} and ψ=12ϑH2+H4.

Solving Eq. (21) yields

(22) 0≤V≤ψ2σ+[V(0)−ψ2σ]exp−(2σt).

It shows that V is uniformly ultimately bounded. From Eq. (17), s1, s2 and H~ are uniformly ultimately bounded. Thus, Theorem 1 is proved.

Optimal control law design

To optimize the tracking performance online, an optimal control law based on ADHDP is developed in this section. Schematic diagram of the proposed closed-loop control system of the UUV is presented in Fig. 2. An action-critic network structure is developed, the actor network is constructed to learn the optimal control scheme τa=[τa1,τa2,τa3]T, and the critic network is constructed to approximate the cost function. Notate s¯=[s(t−Δt),s(t)]T with s=[s11,s12,s13,s21,s22,s23]T. Δt denotes the uniform time interval.

Figure 2 Schematic diagram of the proposed closed-loop control system of the UUV.

Herein, the cost function is formulated as follows:

(23) P(s¯(t),τ(t))=∑c=1∞βc−1Y(s¯(t+bΔt),τa(t+bΔt))

where 0<β≤1 is a discount factor. Y(s¯(t),τa(t))=s¯TFs¯+τaTLτa represents the utility function. Herein, F∈R12×12 and L∈R3×3 are positive-definite design matrices.

The Bellman equation is as follows (Liang, Xu & Zhang, 2023):

(24) P∗(t−Δt)=minτa(t−Δt)⁡[Y(t)+βP∗(t)]

The goal of this section is to make the system output η1 can track the desired trajectory ηd in an optimal manner, while minimizing the cost function.

1) Critic network:

A multilayer perceptron with a three-layer network containing input, hidden, and output layers is introduced to approximate the cost function P(t). The input vector is Ic=[s(t−Δt),s(t),τa]. Ich is the number of hidden nodes. ωcij(1)(t) and ωcj(2)(t) denote the weights of critic network, where i=1,…,15 and j=1,…,Ich. The hyperbolic tangent function Ξ(t)=[(1−e−t)/(1+e−t)] is introduced in this work. Here, pcj(t) and qcj(t) are the intermediate variables. Further, one can derive the approximation P^(t) as

(25) pcj(t)=∑i=112ωcij(1)(t)s¯1i(t)+∑i=13ωc(i+12)j(1)(t)τai(t)

(26) qcj(t)=Ξcj(pcj(t))=1−e−pcj(t)1+e−pcj(t)

(27) P^(t)=ωc(2)(t)Ξc(t)=∑j=1Ichωcj(2)(t)qcj(t).

2) Action network:

A multilayer perceptron with a three-layer network containing input, hidden, and output layers is introduced to approximate the optimal control law τa. The input vector of action network is s¯, and the output vector of action network is τa. Iah is the number of hidden nodes. ωaκn(1)(t) and ωank(2)(t) denote the weights of action network, where κ=1,…,12, n=1,…,Iah and k=1,…,3. Here, pan(t) and qan(t) are the intermediate variables. Further, we have

(28) pan(t)=∑κ=112ωaκn(1)(t)s¯1κ(t)

(29) qan(t)=Ξan(pan(t))=1−e−pan(t)1+e−pan(t)

(30) τak(t)=∑n=1Iahωank(2)(t)qan(t).

3) Adaptation of the critic network:

The critic network’s error function can be described as:

(31) Oc(t)=βP^(t)−[P^(t−Δt)−Y(t)]

Hence, the critic network’s objective function for weights update can be defined as:

(32) Ec(t)=12Oc2(t).

A gradient descent algorithm is applied to update the weight law. Here, ωcij(1)(t+Δt)=ωcij(1)(t)+Δωcij(1)(t), ωcj(2)(t+Δt)=ωcj(2)(t)+Δωcj(2)(t). According to Eqs. (28)–(30), the weight updating laws Δωcij(1) and Δωcj(2) of critic network can be calculated using the chain derivation rule

(33) Δωcij(1)=−μc∂Ec(t)∂P^(t)∂P^(t)∂qcj(t)∂qcj(t)∂pcj(t)∂pcj(t)∂ωcij(1)(t)=−μcβOc(t)ωcj(2)(t)[12(1−Ξcj2(t))]Ici(t)

(34) Δωcj(2)=−μc∂Ec(t)∂P^(t)∂P^(t)∂ωcj(2)(t)=−μcβOc(t)Ξcj(t)

where μc is the learning rate.

4) Adaptation of the action network:

The action network can be adjusted by backpropagating the error between the desired value and the approximate value of the critic network.

Herein, we have

(35) Oa(t)=P^(t)−P(t)

(36) Ea(t)=12Oa2(t)

Similarly, a gradient descent algorithm is applied to update the weight law of the action network. Here, ωaκn(1)(t+Δt)=ωaκn(1)(t)+Δωaκn(1)(t), ωank(2)(t+Δt)=ωank(2)(t)+Δωank(2)(t). According to Eqs. (31)–(33), the weight updating laws ωaκn(1)(t) and Δωank(2) of action network can be calculated using the chain derivation rule:

(37) Δωaκn(1)=−μa∂Ea(t)∂P^(t)[∂P^(t)∂τak(t)]T∂τak(t)∂qan(t)∂qan(t)∂pan(t)∂pan(t)∂ωaκn(1)(t)=−μaOa(t)(∑j=1Ichωcj(2)(t)[12(1−Ξcj2(t))])×(∑i=13ωc(i+12)j(1)(t))ωank(2)(t)[12(1−Ξan2(t))](∑κ=112s¯1κ(t))

(38) Δωank(2)=−μa∂Ea(t)∂P^(t)∂P^(t)∂τak(t)∂τak(t)∂ωank(2)=−μaOa(t)Ξan(t)(∑j=1Ichωcj(2)(t)[12(1−Ξcj2(t))])×∑i=13ωc(i+12)j(1)(t)

where μa is the learning rate.

Remark 1 As a matter of convenience, only the output layer weights of the critic–action network are tuned during the learning process, whereas the weights in Eqs. (25) and (28) are randomly initiated. As described by Liang, Xu & Zhang (2023), as the number of NN’s hidden nodes increases, the NN’s approximation error is able to converge to a value that is adequately small.

Stability analysis

The optimal control law based on ADHDP is developed to optimize the tracking performance online. It can be obtained that actual control input signal is compound of the robust adaptive NN control law and optimal control law. The stability of robust adaptive control has been presented in Theorem 1. The ADHDP technique is looking for an optimal control action while approximating the Bellman equation with the critic network.

The ideal weights ωc∗ and ωa∗ for critic and action networks are bounded.

(39) ∥ω∥c∗≤ωcm,∥ωa∗∥≤ωam

where ωcm and ωam are positive constants.

In the light of Eqs. (27) and (30), we have:

(40) P^(t)=ωc(2)(t)TΞc(t)

(41) τa=ωa(2)(t)TΞa(t)

where ω~c(2)=ωc(2)−ωc∗ and ω~a(2)=ωa(2)−ωa∗ are the estimation error of ωc(2) and ωa(2), respectively.

Theorem 2 Consider the updating law Eq. (38) for actor network weights, and the updating law Eq. (34) for critic network weights, and the utility function Y(t) is assumed as a bounded positive semidefinite function. Then, the estimation error ω~c(2) and ω~a(2) are bounded.

Proof: Select the following Lyapunov function candidate as:

(42) LV(t)=LV1(t)+LV2(t)

where LV1(t)=1μctr(ω~c(2)(t)Tω~c(2)(t)), LV2(t)=1γμatr(ω~a(2)(t)Tω~a(2)(t)) with γ being a design constant.

The first difference of LV(t) can be written as:

(43) ΔLV(t)=ΔLV1(t)+ΔLV2(t).

Here, ΔLV1(t) and ΔLV2(t) can be calculated as

(44) ΔLV1(t)=1μctr(ω~c(2)(t+Δt)Tω~c(2)(t+Δt)−ω~c(2)(t)Tω~c(2)(t))

(45) ΔLV2(t)=1γμatr(ω~a(2)(t+Δt)Tω~a(2)(t+Δt)−ω~a(2)(t)Tω~a(2)(t)).

From Eq. (34), ϖ~c(2)(t+Δt) can be get:

(46) ω~c(2)(t+Δt)=ω~c(2)(t)+Δω^c(2)(t)=−μcβΞc(t)[βωc(2)(t)TΞc(t)+Y(t)−ωc(2)(t−Δt)TΞc(t−Δt)]T+ω~c(2)(t).

According to Eqs. (44) and (46), we have:

(47) ΔLV1=−2βδc(t)[Y(t)−ωc(2)(t−Δt)TΞc(t−Δt)+βωc(2)(t)TΞc(t)]T+μcβ2∥Ξc(t)∥2∥βωc(2)(t)TΞc(t)+Y(t)−ωc(2)(t−Δt)TΞc(t−Δt)∥2

where δc(t)=ω~c(2)(t)TΞc(t).

Defined −2βδc(t)[Y(t)−ωc(2)(t−Δt)TΞc(t−Δt)+βωc(2)(t)TΞc(t)]T=A. The first term in Eq. (47) can be defined as:

(48) A=∥βωc(2)(t)TΞc(t)+Y(t)−ωc(2)(t−Δt)TΞc(t−Δt)−βδc(t)∥2−∥βωc(2)(t)TΞc(t)+Y(t)−ωc(2)(t−Δt)TΞc(t−Δt)∥2−∥βδc(t)∥2.

Substituting Eqs. (48) into (47), we have:

(49) ΔLV1(t)=−β2(1−μcβ2||Ξc(t)||)×||β−1Y(t)−β−1ωc(2)(t−Δt)TΞc(t−Δt))+ωc(2)(t)TΞc(t)||2+||βωc∗Ξc(t)−ωc(2)(t−Δt)TΞc(t−Δt)+Y(t)||2−β2||δc(t)||2.

From Eq. (38), we have:

(50) ω~a(2)(t+Δt)=ω~a(2)(t)+Δω^a(2)(t)=ω~a(2)(t)−μaΞa(t)ωc(2)(t)TN(t)[ωc(2)(t)TΞc(t)]T

According to Eqs. (45) and (50), we have:

(51) ΔLV2(t)=1γ1tr(−2δa(t)ωc(2)(t)TN(t)[ωc(2)(t)TΞc(t)]T+μa∥Ξa(t)∥2∥ωc(2)(t)TN(t)∥2∥ωc(2)(t)TΞc(t)∥2)

where δa(t)=ω~a(2)(t)TΞa(t).

The last term in Eq. (51) can be further calculated as:

(52) −2δa(t)ωc(2)(t)TN(t)[ωc(2)(t)TΞc(t)]T=∥ωc(2)(t)TΞc(t)−ωc(2)(t)TN(t)δa(t)∥2−∥ωc(2)(t)TΞc(t)2∥−∥ωc(2)(t)TN(t)δa(t)∥2.

Substituting Eqs. (52) into (51) yields

(53) ΔLV2(t)=1γ1tr(∥ωc(2)(t)TΞc(t)−ωc(2)(t)TN(t)δa(t)∥2+μa∥Ξa(t)∥2∥ωc(2)(t)TN(t)∥2∥ωc(2)(t)TΞc(t)∥2−∥ωc(2)(t)TΞc(t)∥2−∥ωc(2)(t)TN(t)δa(t)∥2)

By using Cauchy–Schwarz inequality, we have:

(54) ∥ωc(2)(t)TΞc(t)−ωc(2)(t)TN(t)δa(t)∥2−∥ωc(2)(t)TΞc(t)∥2≤4∥δc(t)∥2+4∥ωc∗TΞc(t)∥2+∥ωc(2)(t)TN(t)δa(t)∥2.

Substituting Eqs. (54) into (53), we have:

(55) ΔLV2(t)≤1γ[−(1−μa∥Ξa(t)∥2∥ωc(2)(t)TN(t)∥2)×∥ωc(2)(t)TΞc(t)∥2+4∥δc(t)∥2+4∥ωc∗TΞc(t)∥2+∥ωc(2)(t)TN(t)δa(t)∥2].

According to Eqs. (49) and (55), we have:

(56) ΔLV(t)=ΔLV1(t)+ΔLV2(t)=−β2(1−μcβ2∥Ξc∥2)×∥ωc(2)(t)TΞc(t)+β−1Y(t)−β−1ωc(2)(t−Δt)TΞc(t−Δt)∥2−(β2−4γ)∥δc(t)∥2−1γ1∥ωc(2)(t)TΞc(t)∥2×(1−μa∥Ξc(t)∥2∥ωc(2)(t)TN(t)∥2)+Ω

where Ω=4γ||ωc∗TΞc(t)||2+1γ∥ωc(2)(t)TN(t)δa(t)∥2 +∥βωc∗TΞc(t)+Y(t)−ωc(2)(t−Δt)TΞc(t−Δt)∥2.

By using Cauchy–Schwarz inequality, we have:

(57) Ω(t)≤(4β2+4γ)∥ωc∗(t)TΞc(t)∥2+1γ∥ωc(2)(t)N(t)δa(t)∥2+4∥Y(t)∥2+2ωc(2)(t−Δt)TΞc(t−Δt)∥2≤(4β2+4γ+2)ω¯c2Ξcm2+4Ym2+1γωcM2Nmω¯am2Ξam2=Dm

where ω¯am, ωcM, Ξcm, Ξam, Ym and Nm are the upper bounds of ω~a(t), ω~c(t), Ξc(t), Ξa(t), Y(t) and N(t), respectively, and ω¯c=max{ωcm,ωcM}.

Choosing μa<[1/(∥Ξa(t)∥2∥ωc(2)(t)M(t)∥2)], ||δa(t)||2>[Dm/(β2−4γ)], μc<[1/(β2||Ξc(t)||2)] and β2−4γ>0, then ΔLV(t)≤0.

According to the Lyapunov stability theorem, the estimation error ω~c(2) and ω~a(2) are bounded.

Remark 2 The actual input signal of the UUV consists of the robust adaptive control law and the optimal control law. The velocity error and trajectory tracking error under the robust adaptive control law are the driving signals of the optimal control law based on ADHDP. In other words, the optimal control law based on ADHDP is inactive and the equivalent output is 0 if the velocity and trajectory are able to track the desired trajectory under the robust adaptive control law.

Remark 3 With respect to the trajectory tracking control law developed in Li et al. (2021) and Yang et al. (2021), manual adjustment of the control law design parameters does not easily guarantee control performance. Once the UUV suffers from the bigger uncertain dynamics and time-varying disturbances, the fixed control law design parameters can’t attain the desirable tracking accuracy. With the incorporation of ADHDP into the control law design, the control law design parameters are updated automatically, thus effectively improving the control performance of the system.

Simulation results

In this section, we perform some simulations on an underactuated UUV to show the effectiveness and superiority of the proposed control scheme. As a comparison, the robust adaptive NN control scheme without the optimal control law based on ADHDP is labeled as “RC”. As a comparison, an advanced kalman filter-Based control scheme based on Vafamand, Arefi & Anvari-Moghaddam (2023) is labeled as “AKF” and the design parameters about the augmented Kalman filter is selected the same as Vafamand, Arefi & Anvari-Moghaddam (2023). The ADHDP-based self-learning control scheme proposed is labeled as the “ADHDP”. The model parameters of the UUV (Liang et al., 2020) are given as follows: m11=47.5kg, m22=94kg, m33=94kg, m44=13.5kg⋅m2, m55=13.4kg⋅m2, d11=13.5kg⋅s−1, d22=45kg⋅s−1, d33=45kg⋅s−1, d44=23.8kg⋅m2⋅s−1, d55=27.2kg⋅m2⋅s−1 and ρg∇GML=8.8.

The unknown disturbance vector is given as d(t)=[5sin⁡(0.15t)+2cos⁡(0.15t),0.5sin⁡(0.15t)+0.5cos⁡(0.15t),0.5sin⁡(0.15t)+0.5cos⁡(0.15t),sin⁡(0.15t)+cos⁡(0.15t),sin⁡(0.15t)+cos⁡(0.15t)]T. The desired trajectory is given as xd=10cos⁡(0.1t), yd=10sin⁡(0.1t) and zd=−10−0.1t.

The initial conditions are chosen as η(0)=[8.5m,1m,−9m,0.1πrad,0rad]T, v(0)=[0m/s,0m/s, 0m/s,0rad/s,0rad/s]T. The design parameters of the robust adaptive NN control law are selected as F=0.3, Γ=0.8, ϑ=0.002, k1=diag(0.2,0.1,0.2), k2=diag(10,10,10).

The design parameters of optimal control law are chosen as β=0.95, F=0.001I12×12, L=0.001I3×3, respectively. The learning rates are selected as μa(0)=0.1 and μc(0)=0.1. The learning rate gradually decreases to the final values μa(∞)=0.005 and μc(∞)=0.005 over time. The number of hidden nodes for the action–critic network are given as Ich=6, Iah=5 and the initial weights of action–critic network are randomly generated within the range [−0.2,0.2]. The maximal iteration numbers are set as na=200 for the action network and nc=200 for the critic network. The completion of internal training is determined by reaching either the maximal iteration numbers. Herein, define the performance metrics as Js1=∫0T(|s11|+|s12|+|s13|)dt, Js2=∫0T(|s21|+|s22|+|s23|)dt.

Figures 3–8 show the simulation results of the proposed control scheme and the comparison control scheme. From Fig. 3, it can be seen that the underactuated UUV can accurately track the 3D spiral drive trajectory by the proposed control scheme. Specifically, it can be seen from Fig. 4 that the proposed control scheme has a smaller tracking error compared to the these two control scheme. It is observed in Fig. 4 that the tracking error under “AKF” is smaller than the tracking error under “RC”. The reason for this is that the main role of Kalman filtering in control is to provide accurate estimates of the state of the system, especially in the presence of noise and uncertainty, and these estimates help to improve the control strategy and hence the stability and performance of the system. As can be seen in Figs. 5, 6, the optimal control part is enacted to minimize the tracking error resulted from the initial state deviation. As the weights are adaptively adjusted, the tracking error and utility function gradually decrease. As the tracking error approaches 0, the utility function decreases to 0 and the weights remain constant. Moreover, the control performance can be assessed through the performance metrics as shown in Fig. 7. Clearly, the performance metric values of the proposed control scheme are smaller than the metric values of the compared control scheme, corroborating the feature that the proposed control scheme is able to optimize the tracking performance. The performance metric values under “AKF” is smaller than under “RC”. The effect of Kalman filtering is further demonstrated in Fig. 7. As shown in Fig. 8, the control input of the proposed control scheme is within a reasonably bounded range.

Figure 3 Desired and actual trajectories of the UUV.

Figure 4 The tracking error of the UUV.

Figure 5 The weights of action network.

Figure 6 Utility function.

Figure 7 The performance metrics.

Figure 8 The actual control input.

Conclusion

In this work, we have proposed a robust self-learning control scheme based on ADHDP to deal with the 3D trajectory tracking control problem of an UUV with uncertain dynamics and time-varying ocean disturbances. By combining the ADHDP optimization control scheme with the robust adaptive control scheme, an adaptive self-learning optimal scheme with online learning function is proposed. The proposed control method requires less model information and is more suitable for the actual situation of the system. In addition, the control parameters can be automatically updated according to the changes of external environment and unknown dynamics. Theoretical analyses as well as the comparison of simulations show that the proposed control scheme has significant effectiveness and superiority.

Supplemental Information

Supplemental Information 1 The model parameters of the UUV and all the design parameters of the control law.

Additional Information and Declarations

Competing Interests

Author Contributions

Data Availability

The authors declare that they have no competing interests.

Chunbo Zhao performed the experiments, analyzed the data, prepared figures and/or tables, authored or reviewed drafts of the article, and approved the final draft.

Huaran Yan conceived and designed the experiments, prepared figures and/or tables, authored or reviewed drafts of the article, and approved the final draft.

Deyi Gao performed the computation work, authored or reviewed drafts of the article, and approved the final draft.

The following information was supplied regarding data availability:

The model is available in the Supplemental File.

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
