# Peer review of "ADHDP-based robust self-learning 3D trajectory tracking control for underactuated UUVs"

_PeerJ Computer Science, doi:10.7717/peerj-cs.2605_

## Round 0.1 · original submission · Major Revisions

I consider the research is well-defined. The article is an original one and aligns well with the journal's aims and scope.
The authors should mention and justify the novelty of the methodology proposed.
I strongly suggest comparing the results with State Observers or a Kalman Filter as a reference.
The paper must be improve according to all observations introduced by reviewers.

Reviewer 1 ·

Basic reporting

Attachment

Experimental design

Attachment

Validity of the findings

Attachment

Additional comments

Attachment

Annotated reviews are not available for download in order to protect the identity of reviewers who chose to remain anonymous.

Reviewer 2 ·

Basic reporting

The article focuses on self-learning control of an Unmanned Underwater Vehicle (UUV) to navigate and address the problem of under-actuation. Here are my comments to strengthen the article:

1. The contribution section of the article is unclear. The authors should reword this section to clearly define the novelty of their work.
2. Since you mention robust control in the presence of disturbances, please cite relevant articles in the introduction and discuss the current research in this area.
3. There are several types of UUVs. Which model type have you chosen for your paper? Include a picture or sketch of your model to clarify and justify your mathematical model.
4. Why is rotation about the x-axis not considered in the mathematical model?
5. Include a section explaining control and stability concerning the UUV.Show the overall control architecture in the form of a figure.
6. In the simulation, plot the disturbance graph as well to make it easier to read and compare the results.
7. It is good that the proposed method is working. However, I strongly suggest comparing the results with State Observers or a Kalman Filter as a reference, as Kalman filters are commonly used in experiments.

Experimental design

There are no experimental results

Validity of the findings

See my comments

Additional comments

The paper needs major improvement. Also, check the grammatical mistakes.

---

## Round 0.2 · Major Revisions

The paper must be improved. My decision is Major Revisions.

Reviewer 2 ·

Basic reporting

Thank you for the answers. However, I still find that the novelty is not clear, and the paper has many spelling and grammatical errors.

I believe the authors rushed to submit the second version. The figures are not readable, and the cited papers are mostly from 2009. Please cite and address the latest research in the field.

For Figure 1, replace it with a picture of the UUV and place the reference frames on top of it. Figures 2 and 3 are also unclear. In fact, please check all the figures for clarity.

Experimental design

No experimental setup

Validity of the findings

The proposed idea shows the effectiveness of the proposed method compared to Kalman Filter, but the figures are not clear and it is not explained well in the text.

Additional comments

-

---

## Round 0.3 · accepted · Accept

The paper was well improved and can be accepted.

Reviewer 2 ·

Basic reporting

No further comments.

Experimental design

N/A

Validity of the findings

No further comments

Additional comments

No further comments